# Assessment of Microbial Contamination in the Infulene River Basin, Mozambique

Clemêncio Nhantumbo [1,*], Nídia Cangi Vaz [2], Mery Rodrigues [3], Cândido Manuel [1], Sífia Rapulua [1], Jéssica Langa [1], Hélio Nhantumbo [1], Dominic Joaquim [1], Michaque Dosse [1], José Sumbana [3,4], Ricardo Santos [5], Silvia Monteiro [5] and Dinis Juízo [1]

1   Faculty of Engineering, Eduardo Mondlane University, Av. de Moçambique km 1.5, Maputo 1102, Mozambique
2   Biotechnology Centre, Eduardo Mondlane University, Av. de Moçambique km 1.5, Maputo 1102, Mozambique
3   Sciences Faculty, Eduardo Mondlane University, Main Campus, Av. Julius Nyerere 3453, Maputo 1102, Mozambique
4   Faculty of Medicine, Eduardo Mondlane University, Av. Salvador Allende no 702, Maputo 1102, Mozambique
5   Instituto Superior Tecnico, Universidade Lisboa, Av. Rovisco Pais, 1049-001 Lisboa, Portugal
*   Correspondence: n_clemencio@hotmail.com or clemencio.d.nhantumbo@uem.ac.mz

**Abstract:** Water microbial contamination is one of the major threats to human health. The study focus is on Infulene River Basin, a urban catchment with mainly informal settlements, with limited water supply and sanitation. In the catchment there are two wastewater treatment plants, one hospital and beer factory located on the banks of the main stream; water from this stream is used for urban agriculture and domestic uses by some dwellers. These factors present a significant health risk from water-borne diseases. At the moment there is limited knowledge about the level of microbial contamination of the different sources of water at the disposal of the communities. Thus, a preliminary study on fecal microbial contamination was conducted targeting the Infulene River and the drainage system from the nearby Maputo city draining into the system, with additional investigation on the drinking water provided by the city water supply company. The quantification of *Total Coliforms (TC)* and *Escherichia coli (EC)* was conducted at several sampling locations. Results were compared with official drinking water standards. Eighty two percent (82%) and 61% of Infulene river water and drainage water samples were positive for *TC* ($10^5$ to $10^9$ NPN/100 mL) and *EC* ($10^5$ to $10^7$ NPN/100 mL), respectively. For drinking water samples, 63% and 23% were positive for *TC* (up to 6000 NPN/100 mL) and *EC* (up to 1000 NPN/100 mL), respectively. Higher microbial contamination was found in neighborhoods with the poorest sanitation and shallow groundwater, i.e., Chamanculo, Xipamanine, Mafalala, Aeroporto and Maxaquene, a situation that was more expressive during the rainy season. Overall, the study confirmed the high vulnerability to microbial contamination of all sources investigated due to poor sanitation and lack of drainage infrastructure. The risks to human health might be even higher considering that contaminated water is used for gardening of vegetable watering and domestic use.

**Keywords:** microbial contamination; river water; total coliforms; *Escherichia coli*; drinking water

## 1. Introduction

Rivers are important sources of water, playing an important role by providing water for domestic and industrial activities, livestock production, agriculture, drinking water and other uses [1,2]. Nevertheless, anthropogenic activities cause water contamination through direct discharge of effluents into the receiving water bodies and streams. Globally, more than 80% of wastewater is discharged directly into the environment without proper treatment, which leads to contamination by pathogenic microorganisms such as bacteria, viruses, protozoa and helminths [3,4].

Microbial pollution in aquatic environments is one of the pressing issues affecting the sanitary state of water sources for domestic water uses, recreational activities and harvesting seafood [5]. Pathogenic organisms are natural livings things of all ecosystems, but microbiological contamination with fecal bacteria from anthropogenic activity is increasingly contributing to the degradation of water sources throughout the rivers and drinking water systems worldwide [4,6–8]. There is an increased body of literature reporting on microbial contaminated waters from human and animal activities affecting river water quality, with prominent impacts on the lower reaches of the cities [9–11].

Assessment of human enteric pathogens in eight rivers used as rural household drinking water sources in the northern region of South Africa showed that the indicator bacteria counts exceeded drinking water quality guideline limits [11]. Another study conducted in Kelani River Basin, Sri Lanka revealed that the entire river was contaminated with total coliform and E. coli bacteria at almost all the sampling locations [10].

Recently, the reliability of fecal indicators as a means of assuring water safety is considered increasingly challenging for the water quality and public health sectors [8]. While in developing countries, water-borne pathogens are the second cause of gastrointestinal diseases of children, where 21% of children do not reach the age of five, with about 2.5 million of deaths per year [12].

In Mozambique, there are many cases of diarrhea associated with consumption of contaminated water, especially after the rainy season, with diarrhea being the third cause of death in infants and young children [13–15]. Among diseases that cause gastroenteritis, cholera is endemic in Mozambique and Maputo with 65% of total cases in the whole country. This situation has nevertheless improved between 2011 and 2017; of the 278 cases, only one fatality was reported [16]. Additionally, the same study revealed the occurrence of EC, TC and *V. cholerae* in drinking water in Maputo [16]. Peri-urban areas with unplanned settlement seem to be much more affected by the problem due to lack of sanitation compounded by poor water supply services.

This study focuses on the Infulene River, a small river of approximately a 20 km stretch. It is located in the Southern part of Mozambique between the cities of Maputo and Matola, flowing into Espirito Santo estuary [17]. The land use of the basin encompasses urban, peri-urban characterized by informal settlements with poor sanitation systems and rural areas. The informal settlements are characterized by low quality domestic water distribution services, with intermittent water supply, together with poor sanitation. In these circumstances, when water supply is interrupted and hydraulic pressure in the distribution system decreases, there is a possibility of ingress of contaminated water into the distribution pipes, especially during the rainy seasons [18]. Additionally, surface runoff contributes to the spreading of contaminants from the informal settlements, thus decreasing the water quality of receiving streams [19].

A study recently published aims to assess the microbiological quality of water sold by street vendors along the main drainage channel in Maputo city, targeting the determination of the antibiotic resistance profile of selected Enterobacteriaceae isolates. The study reported that 88% of samples analyzed were positive for fecal coliforms and 66% were positive for *EC* on bottled water. These results where compared to samples of tap water, and 64% were found positive for fecal coliforms and 28% were positive for *EC* [20]. Additionally, of the 44 selected Enterobacteriaceae isolates from water samples (28 isolates of EC and 16 isolates of Klebsiella spp.), 45.5% were not susceptible to the beta-lactams ampicillin and imipenem, 43.2% to amoxicillin, and 31.8% to amoxicillin/clavulanic acid. Regarding non-beta-lactam antibiotics, there was a high percentage of isolates with tolerance to tetracycline (52.3%) and azithromycin (31.8%) [20]. Another study on microbial contamination in Maputo Bay, in the part of Maputo city center which has better sanitation infrastructure, demonstrated that pathogens are common at some sites [21]. While both studies demonstrated prevalence of contaminated water used in some areas of the city, none of them have been specific in terms of determining the origins of the waters analyzed. For example, the first study was more concerned in comparing the quality of water from street vendors without examining the

sources, while the second study evaluated the efficacy of the drainage system in eliminating pollution of the receiving waters at the Maputo Bay.

This study aims to reduce the knowledge gap on the status of microbial contamination of the Influene system, as well as detect the patterns of spreading of contaminated water during rainy season in a urban catchment [22]. To this end, an integrated study covering the drainage system, natural stream, as well as drinking water is conducted to map the potentially spreading of microorganisms during the rainy season.

## 2. Methods

### 2.1. Study Area

#### 2.1.1. Infulene River

The Infulene River has a total area of 147 km$^2$ and its longest stream is approximately 20 km. The basin boundaries are Marracuene District to the north, Maputo Bay to the south, Maputo City to the east and Matola City to the west, Figure 1. The climate of the area shows a clearly distinguishable rainy season (October to March) and a dry season (April–September). The annual rainfall ranges from 400 to 1000 mm. The mean annual temperature varies from 22 °C to 29 °C. The relative air humidity varies between 67.3% and 80.5%.

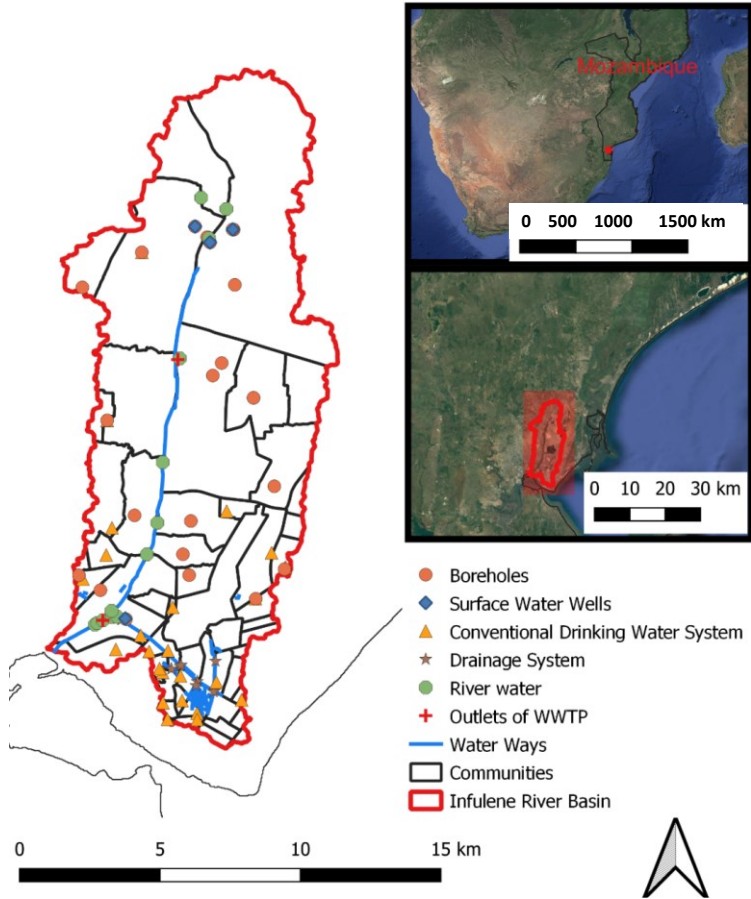

**Figure 1.** Study area and sampling points.

#### 2.1.2. Drainage System

In part of the basin there is a sanitation infrastructure designed to safely collect domestic and industrial wastewater and rainwater to discharge points. This minimizes the risk of flooding and prevents the pollutants from reaching the receiving environments [23,24]. In the Municipality of Maputo, during the rainy season, the wastewater and rainwater drainage networks are interconnected, constituting a combined sewer system. The high

population density in the informal settlements that characterize the peri-urban areas, coupled with lack of sanitation leads to direct discharge of domestic wastewater into the drainage ditches, causing not only microbiological contamination, but also attracting insects and a bad smell. This situation is further exacerbated by the high groundwater table that limits infiltration and makes the construction of pit latrines difficult [25].

### 2.1.3. Drinking Water Systems

This study conducts a comprehensive assessment of water used by the communities from different sources in order to identify the main macrobiotic contamination risks associated with it. Three main drinking water sources are used in the study area, Figure 2. The conventional city water distribution system serves two distinct areas: (a) the consolidated urban area with sanitation system and (b) the informal settlement with both low-quality water distribution system and sanitation. There are also small, reticulated water distribution systems in suburbs that tap into groundwater using shallow water wells.

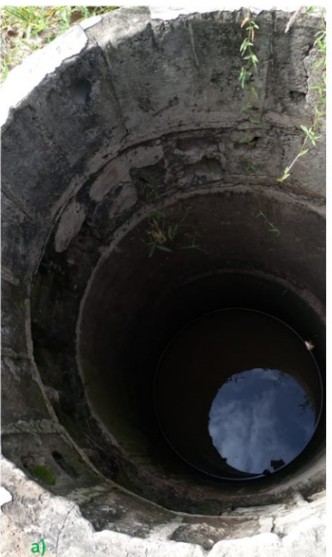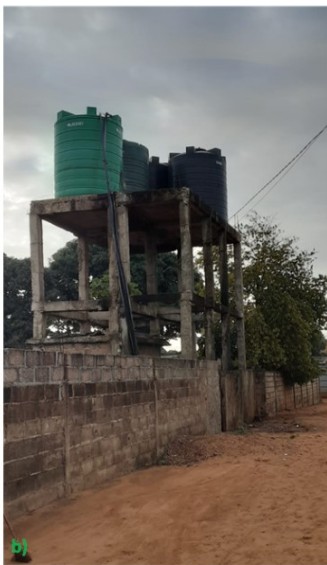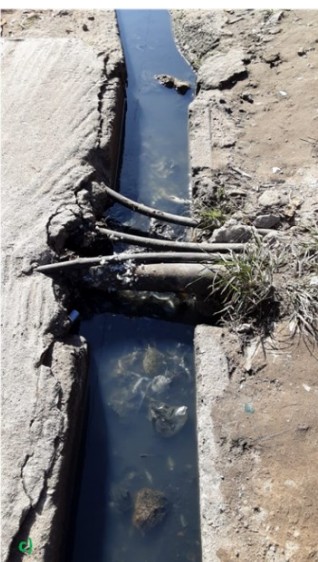

**Figure 2.** Group II—drinking water sources (**a**) Shallow water well, (**b**) small reticulated water supply systems that relay on groundwater, (**c**) poor water distribution and drinking water distribution system in informal settlements.

The consolidate city center is served by a water system that includes a conventional water treatment process, where the chlorination is the final step, aiming to disinfect and prevent re-contamination post treatment. Nevertheless, in the informal settlements the sanitation system is poor, where latrines are flooded during rainy season contaminating surface runoff, with a high probability of sewage water entering the pipes at the time of low pressure, or following a water supply pipe burst, Figure 2c.

In peri-urban areas and informal settlement, there are small pipe water distribution systems that tap into the groundwater through conventional boreholes and store water in elevated tanks for distribution to nearby unserved households or those experiencing low quality of services from the main water system, Figure 2b. In the majority of cases, water is distributed to the final consumers without treatment. This leads to a high probability of water quality degradation in tanks, the water distribution system or even at the household storage of the final user.

In some of the areas, because of financial limitations to construct deep boreholes, dwellers dig shallow large diameter water wells, tapping into the phreatic groundwater aquifer system, Figure 2a. The water is collected using small buckets tied with a rope and carried in 20 L containers to final consumers where it is stored and used without treatment.

In this case, the water is more vulnerable to contamination both at the unprotected well and at the household storage.

### 2.2. Sampling

For the current study, the sampling campaign was conducted between 25 July and 6 August 2022, coinciding with the dry season. Seventy-nine (79) sampling points were selected within the river basin, covering the main pollution sources and water abstraction for agriculture and domestic uses, as well as drinking water sources, Figure 1. The sampling points encompass 13 points on the main river streams, 2 points at the outlets of wastewater treatment plants, 13 points along the stormwater drainage system, and 51 points to evaluating drinking water at the end user. Sampling points for drinking water were selected to gather comparative water quality information for the three main drinking water source used by communities as described above.

Water samples were collected in duplicate at a depth of 10–15 cm and stored in 1-L sterile plastic containers. Thereafter, the samples were refrigerated and transported to the Environmental Engineering Laboratory in the Chemical Engineering Department at Eduardo Mondlane University and kept at a temperature of about 4 °C, for no longer than 24 hours for further analyses.

### 2.3. Physicochemical and Environmental Parameters

Water quality parameters including temperature, conductivity, dissolved oxygen (DO), turbidity and pH of the river water and drinking water were measured in situ using a multi-parameter monitoring instrument (Hach HQ40d, Hach, Loveland, CO, USA). The equipment was calibrated prior to use following the internal standard reference materials, based on the manufacturer's instructions to ensure detection precision.

### 2.4. Laboratory Analysis

The laboratory analysis focused on the most relevant chemical and biological environmental parameters for this study. For chemical parameters, the chemical oxygen demand (COD), nitrates, and phosphates where considered, while biological parameters included Total Coliforms (TC) and *Escherichia coli* (EC).

For measuring phosphates, a total phosphorus test (USEPA PhosVer 3® (Ascorbic Acid), Method 8048 Powder Pillows or AccuVac® Method, Hach, Johannesburg, South Africa) was used. For nitrates, a Cadmium Reduction Method was used (Method 8039 Powder Pillows; AccuVac® Ampuls, Hach, Johannesburg, South Africa). While for COD, a TNT 822 Chemical Oxygen Demand test was used (TNTplus®—Method 8000, Hach, Johannesburg, South Africa). All the results were measured using a DR9000 spectrophotometer (Hach, Johanesburg, South Africa) [26].

The Total Coliforms (TC) and *Escherichia coli* (EC) where determined using the Colilert-18 Test, Product Number: 98-27164-00, Cat. No WP100I-18 (IDEXX, Westbrook, ME, USA), following the manufacturer's instructions. One hundred ml (100 mL) of each water sample was incubated at 35 °C for 18 h and the number of positive wells were converted into a most-probable number (MPN) using the MPN table.

Parameters values were compared with standard threshold values described in Government Decree No. 18/2004 approving the Regulation on Environmental Quality and Effluents' Emissions [27].

### 2.5. Data Analysis

Data was analyzed using statistical and GIS tools. The statistical methods were employed using excel to determine the statistical properties of the data, specifically the average, the errors and variance where average values of measured parameters were calculated and compared with standards and the associated errors and variabilities were further analyzed. The GIS methods were employed using the QGIS Desktop 3.20 software [28],

where spatial distribution of the most relevant parameters was analyzed to identify more vulnerable areas.

## 3. Results and Discussion

### 3.1. Characterization of the Area: Point Pollution Sources

Possible point and diffuse pollution sources can be found in the Infulene River Basin, Figure 3. The possible point pollution sources are two wastewater treatment plants, one hospital, one paper manufacturing company, and one beer factory. The likely diffuse pollution sources are urban agriculture—that is developed on the riverbanks, drainage and wastewater from the dense urban areas—and suburbs (informal settlements) with a deficient sanitation system, Figures 3 and 4. These activities create conditions for possible contamination by microorganisms, nutrients, and organic matter.

The use of a combined system that mixes wastewater and stormwater, such is the case of Maputo, is the most well-known reason of spreading of pathogens, such as bacteria, viruses, and protozoa. Additionally, certain microorganisms adapt and reproduce in the drainage systems [29]. The presence of these contaminants in the urban drainage system can have environmental consequences for the aquatic biota present in the receiving body and for human health [23].

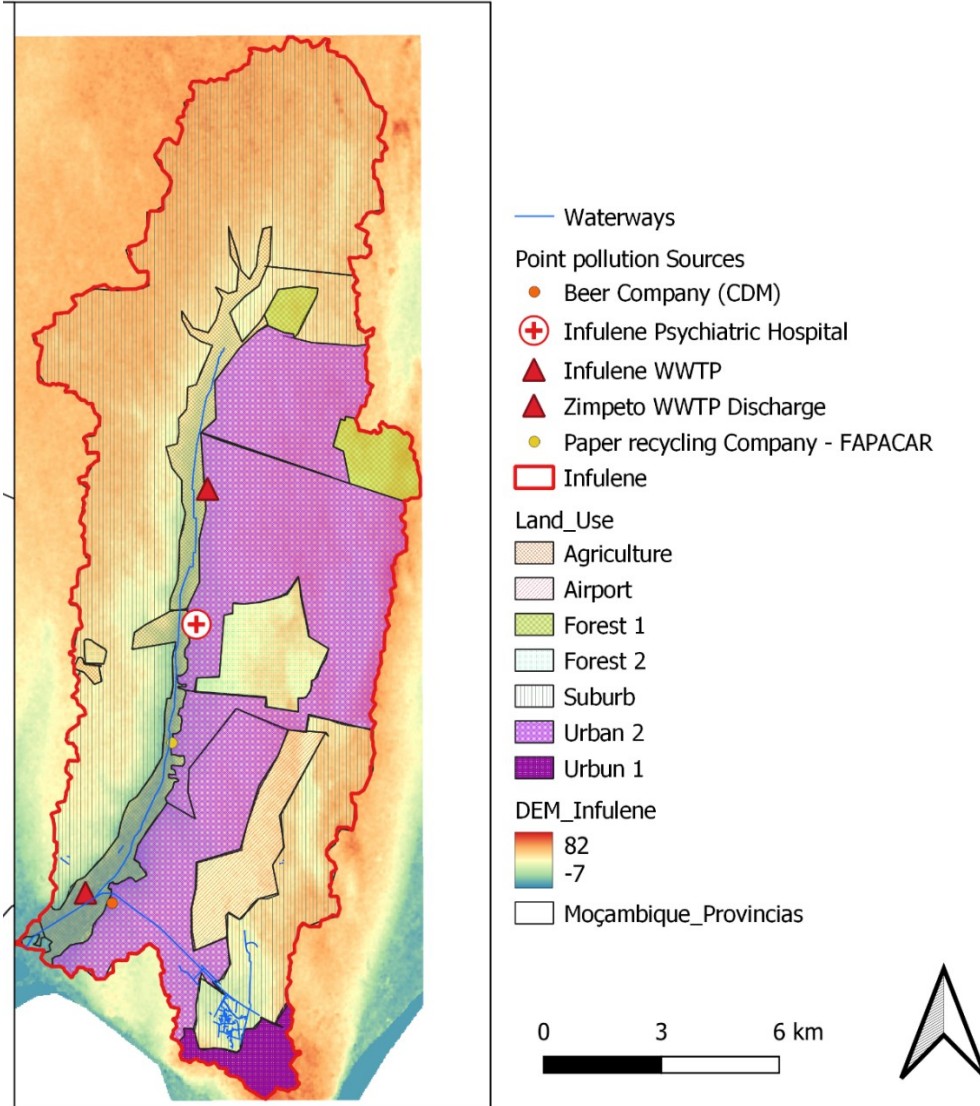

**Figure 3.** Main water point and defuse pollution sources identified.

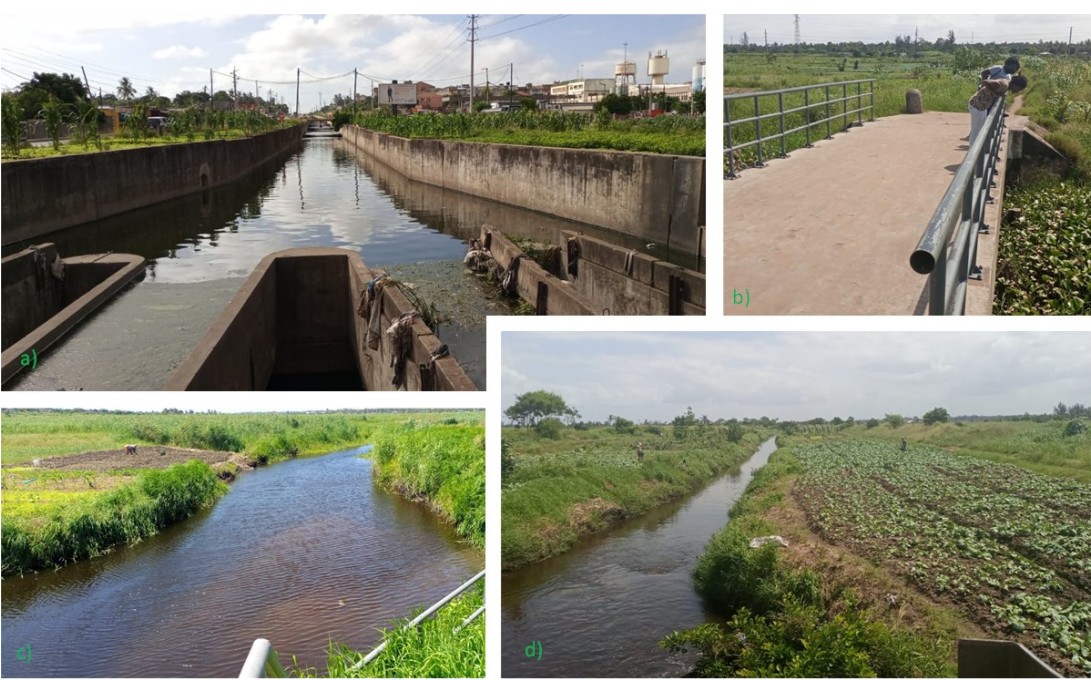

**Figure 4.** Stream and drainage water: (**a**) discharge of drainage water into river stream, (**b**) stream blocked due to excessive growth of vegetation, (**c**) last monitoring point after all target pollution sources, and (**d**) urban agriculture.

*3.2. Water Quality Parameters*

The results of water quality parameters analyzed are divided in two groups: (1) Group I that includes two outlets of wastewater treatment plants, drainage, and river water and (2) Group II that includes drinking water from different sources available in the area, i.e., water from boreholes, conventional distribution system and shallow water wells.

3.2.1. Group I—Outlet of Wastewater Treatment Plants, Drainage, and River Water

For the 28 sampling points of river water, drainage system, and the two outlets of wastewater treatment plants, the average values of water quality parameters are temperature [22.3 (±0.4) °C], pH [7.07 ± 0.05], DO [4.4 (±2.5) mg/L], electrical conductivity [1218 (±411) μS/cm], and COD [75 (±150) mg/L], Figure 5. The results of river water, drainage water, and outlets from the wastewater treatment plants were further analyzed separately. The temperature and pH values were similar for the three sources of water, Figure 5a,c,e. The DO was low for the river water and outlets of wastewater treatment plant, demonstrating a low aeration rate in the river and a heavy load of organic matter in the outlets of wastewater treatment plant, confirmed by the values of COD, Figure 5a,c. In the drainage water, the DO was high which is explained by a high aeration rate because the values of COD were high demonstrating high concentration of organic matter, Figure 5e.

For the case of Infulene River Basin, where agriculture takes place in the margins of the stream, the temperature is within the suitable temperature range, with the optimal range of water for irrigation between 20–30 °C [30]. The pH of the Infulene river is also within the standards, which is 7.25 for natural waters and it may vary between 6.5 to 8.5 [31]. The DO is not in the normal range, (3.2 mg/L), as the level of DO usually required for most of the fish population is between 5 to 6 ppm [31]. Additionally, the value of COD is above the standard, which is 4.0 mg/L [31].

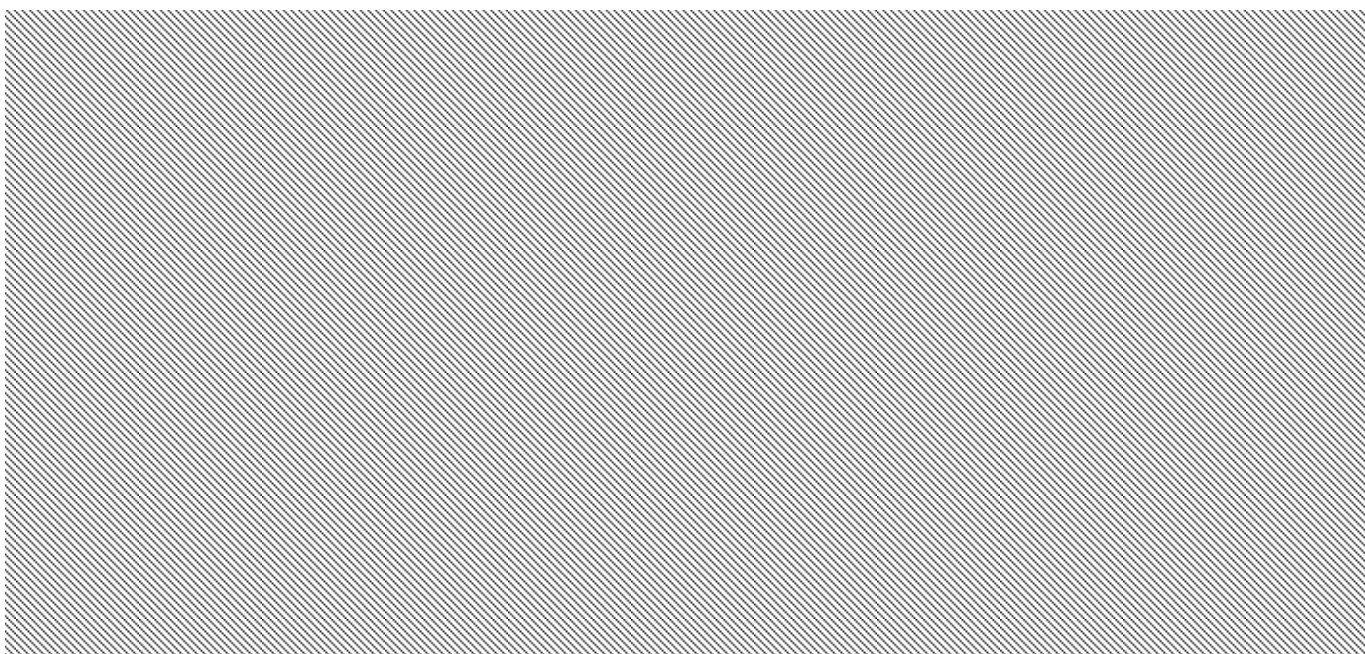

**Figure 5.** Results of water quality parameters from Group I: (**a**) Standard water quality parameters, including Dissolved oxygen (DO), Chemical Oxygen Demand (COD), nitrates and phosphates in river water; (**b**) Biological parameters in river water; (**c**) Standard water quality parameters, including DO, COD, nitrates and phosphates in the outlets of wastewater treatment plants (WWTPs); (**d**) Biological parameters in the outlets of WWTPs; (**e**) Standard water quality parameters, including DO, COD, nitrates and phosphates in the drainage water; (**f**) Biological parameters in the in the drainage water.

The average nutrients concentration in Group I waters are 7.8 ($\pm$6.4) mg/L and 1.7 ($\pm$2.1) mg/L] for nitrates and phosphates, respectively. Considering only the nitrogen and phosphorus within the nitrate and phosphate ions, the average concentration of nitrogen and phosphorus are 1.8 mg/L and 0.54 mg/L, falling in the eutrophic level by being above 1.5 mg/L and 0.075 mg/L for nitrogen and phosphorus, respectively [32]. However, the average concentration of nitrogen and phosphorus, in the river water only, considering the nitrogen and phosphorus within the nitrate and phosphate leads to better results, which are 1.2 mg N/L and 0.35 mg P/L.

The TC [56,000,000 $\pm$ 161,000,000 MPN/100 mL] and EC [4,300,000 $\pm$ 13,000,000 MPN/100 mL] exhibit both high values and variability. Thus, the parameters were considered for spatial variability analyses of water quality of Group I. Separate analysis of river water, outlets from the wastewater treatment plant, and drainage water demonstrated lower concentration of TC and EC in river water samples and higher concentrations in the outlets of wastewater treatment plants and drainage water, Figure 5b–d.

Additionally, from the 28 samples of river water and drainage water analyzed, 82% and 61% of samples were positive for TC and EC, respectively. Sixty seven percent (67%) and twenty seven percent (27%) of the river water samples were positive for TC and EC, respectively, while all samples of drainage water and the outlets from the wastewater treatment plants were positive for both TC and EC.

### 3.2.2. Group II—Water from Boreholes, the Conventional Drinking Water Distribution System, and Shallow Water Wells

The results from the 51 drinking water sampling points exhibit a similar pattern as the results from the river stream and drainage system, with a temperature of [24.1 ($\pm$0.2) °C], pH [7.6 $\pm$ 2.5], DO [7.1 ($\pm$2.5) mg/L], electrical conductivity [612 ($\pm$364) $\mu$S/cm], and turbidity [1.34 $\pm$ 1.30] which are within standard values, as the standard values for drinking water are (Temperature = not defined, pH = 6.5 to 8.5, DO = not defined, electrical conductivity = 50 to 2000 $\mu$S/cm and turbidity = 5 NUTs) [26]. Nevertheless, turbid-

ity exhibited a larger variability, Figure 6. The TC [541 (±1523) MPN/100 mL] and EC [43 (±183) MPN/100 mL] exhibit both high values that exceed the standards for drinking water [27]. Thus, these two parameters will be used for the spatial variability analyses of drinking water quality.

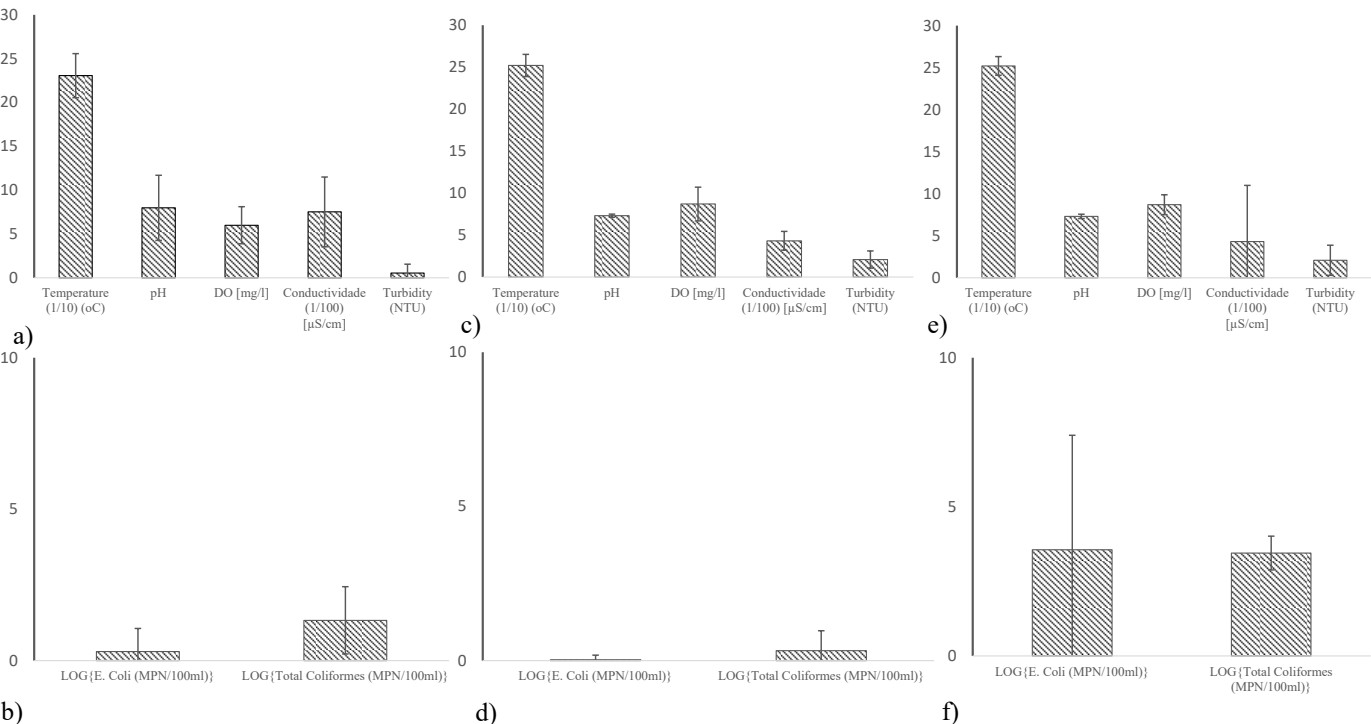

**Figure 6.** Results of water quality parameters from Group II—drinking water sources: (**a**) Standard water quality parameters, including Dissolved oxygen (DO) and Turbidity in the boreholes water; (**b**) Biological parameters in the boreholes water; (**c**) Standard water quality parameters, including DO and Turbidity in the samples collected from the conventional drinking water system; (**d**) Biological parameters in the samples collected from the conventional drinking water system; (**e**) Standard water quality parameters, including DO and Turbidity in the shallow water wells; (**f**) Biological parameters in the shallow water wells; Spatial Analysis.

Separate analysis was done for samples of water collected from boreholes, the conventional drinking water system, and shallow water wells. The results of standard parameters (temperature, pH, electrical conductivity, and turbidity) did not exhibit considerably different values for the three classes of sources of water, Figure 6a,c,e. Nevertheless, the concentration of TC and EC were considerably low for samples of water collected in the conventional drinking water system compared to the concentration of TC and EC in samples collected from boreholes and shallow water wells, Figure 6b,d.

Similarly, from the 51 samples of drinking water analyzed, 63% and 23% were positive for TC and EC, respectively. The discrimination between the different sources of drinking water showed that for boreholes and water samples, 92% and 38% of samples were positive for TC and EC, respectively; for conventional water distribution systems 29% and 8% were positive for TC and EC, respectively; and for shallow water wells, 100% of water samples were positive for TC and 50% of water samples were positive for EC.

When analyzing the occurrence of TC within the river basin, considering Group I water sources, it was observed that the concentration is high along the drainage system with considerably higher concentration at the outlet of the drainage system, Figure 7. TC were ubiquitous in the river and drainage system waters, with extremely high concentrations. High concentrations of TC were also registered from residual waters, in particular, at the

outlets of wastewater treatment plants (the discharge of Zimpeto wastewater treatment plant and discharge of the wastewater from the beer company).

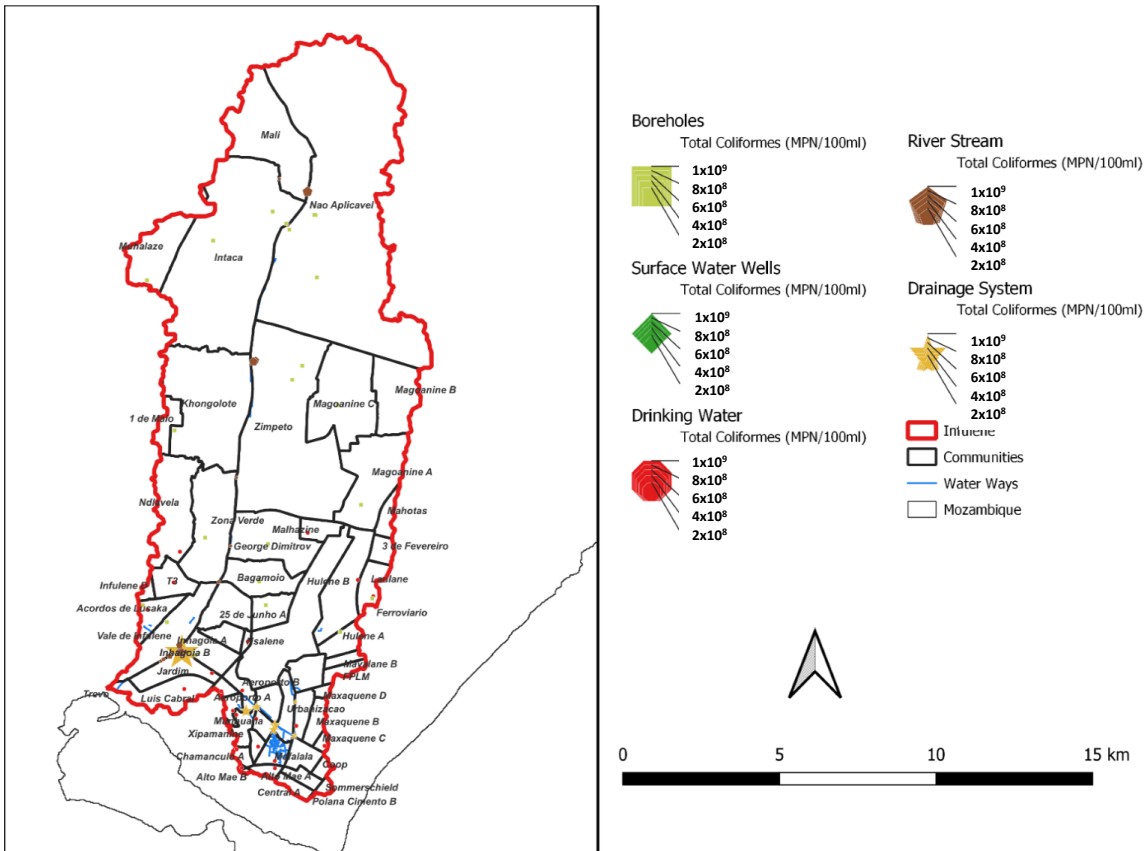

**Figure 7.** Spatial analysis of concentration of TC within the Infulene River Basin, considering both the river water, drainage water; and drinking water.

For better visualization, data on TC from the river and drainage system is further presented separately from the drinking water, Figure 8. A high concentration of TC, around $10^9$ MPN/100 mL, are observed at the outlet of the wastewater coming from the beer company. The drainage system also transports a high concentration of TC with the drainage from Chamanculo and Xipamanine contributing more with TC, followed by the drainage from Mafalala, Aeroporto, and Maxaquene which are areas characterized by informal settlements with very limited sanitation system (Figures 7 and 8a). However, the concentration of TC along the river stream before the interception with the drainage system is low compared to the water from the drainage system, except at the two points, the point immediately before the interception with the drainage system and the interception with the outlet of Zimpeto wastewater treatment plant that exhibited values around $10^7$ NPN/100 mL.

When analyzing the drinking water separately, high concentrations of up to 6000 MPN/100 mL are observed for TC at the end users in water from boreholes and shallow wells. The occurrence of TC in boreholes does not show any spatial pattern (Figure 9b). Although being present in 29% of the samples, water from conventional water samples showed lower concentrations of TC (up to 300 NPN/100 mL).

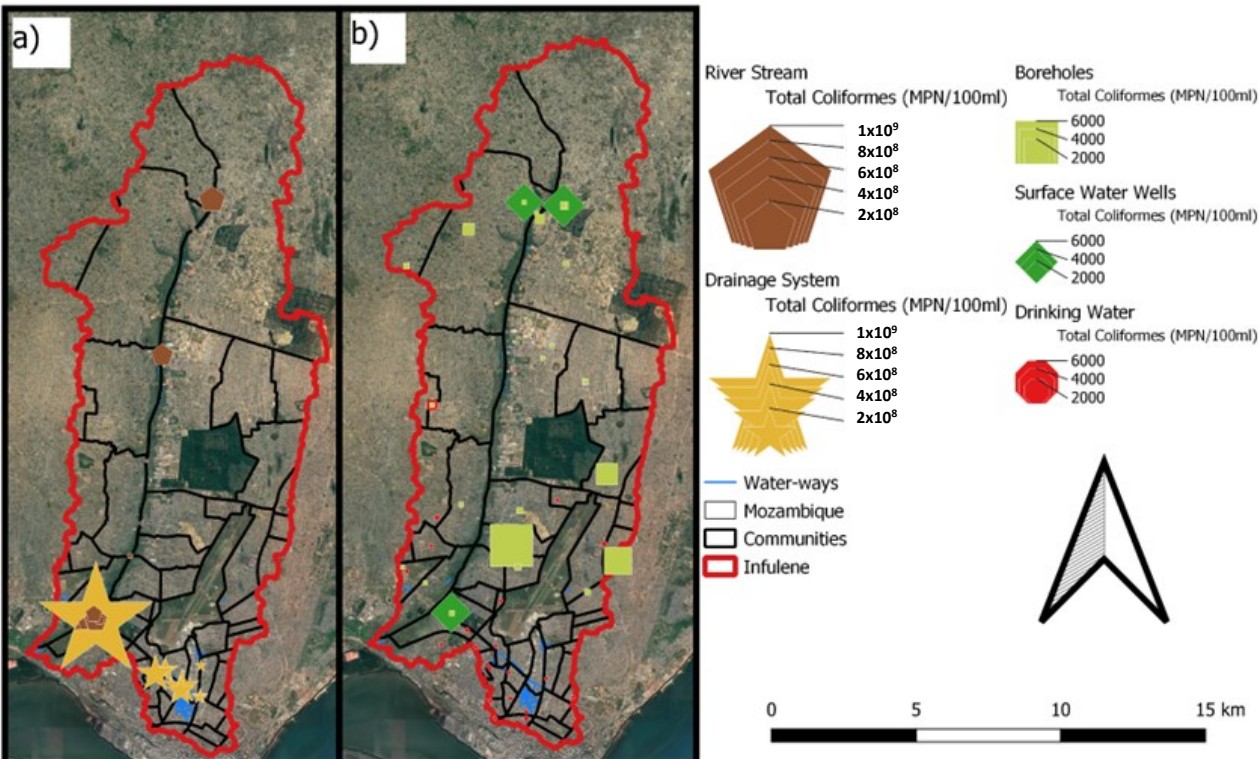

**Figure 8.** Spatial analysis of concentration of TC, considering separately (**a**) the river water with drainage water and (**b**) the drinking water.

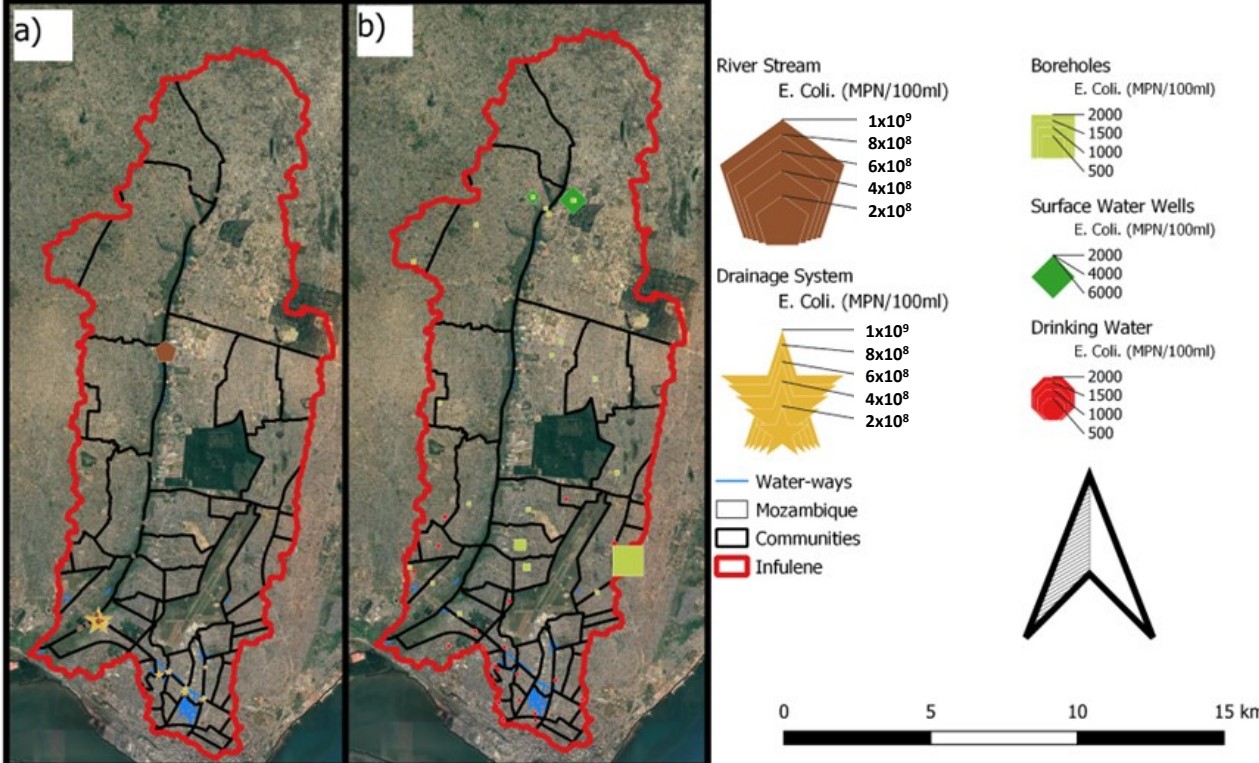

**Figure 9.** Spatial analysis of concentration of *E. coli*, considering separately (**a**) the river water with drainage water and (**b**) the drinking water.

As expected, EC was found at lower concentrations than TC in most of the sampling sites following the same distribution as TC. The concentration observed at the outlet of the wastewater from the beer company is around $10^7$ MPN/100 mL and the drainage system exhibits values between ($10^5$ to $10^7$ MPN/100 mL), Figure 8. The points immediately before mixing with the drainage water and near the outlet of the Zimpeto wastewater treatment plant exhibit concentrations between $10^6$ and $10^7$ MPN/100 mL, respectively.

The occurrence of TC and EC in the boreholes and shallow water wells is explained by the lack of disinfection, poor sanitation, and shallow water table in the areas where those sources are used for drinking water. The borehole water normally is contaminated during storage and utilization while the water from the shallow wells can be contaminated directly at the source. The water from the conventional water distribution system is disinfected at the drinking treatment plant and it carries residual chlorine that is expected to continue disinfecting the water through the distribution system. However, the presence of TC and EC that is observed in some of the sample in some places at the end user might be caused by the depletion of residual chlorine. This fact could be associated with a situation whereby there is poor sanitation, pipes that allow ingress of external contaminated water, as well as the intermittent water supply that characterizes the water supply in informal settlements of Chamanculo, Xipamanine, Mafalala, Aeroporto, and Maxaquene. A previous study on the Maputo Water Supply has shown that drinking water from treatment plants with residual chlorine and without EC gets contaminated in the distribution system when the water supply is intermittent [33].

Another study conducted in Bairro dos Pescadores revealed that the concentration of TC in Costa do Sol, Ponta Vermelha, and Zambi were 465 MPN/100 mL, 225 MPN/100 mL, and 136 MPN/100 mL, respectively [21]. These results, although from a different part of the city, found similar concentrations of TC in drinking water. Additionally, a study conducted in the city of Maputo analyzing drinking water, showed the existence of microbiological contamination on water sold by street vendors along the main drainage channel. This study also included the antibiotic resistance profile of selected Enterobacteriaceae isolates [20]. In the present study, high levels of fecal contamination on tap water samples and even higher in small water suppliers that rely on boreholes were observed. Similar results on TC and EC were observed on studies about river water quality around the world, involving other developing countries [10].

With the finding in the current study, there is a pressing need to identify potential solutions on how to improve the microbial quality of drinking water within the Infulene River Basin. Still, some common solutions should be applied such as introducing disinfection for the small water suppliers that rely on boreholes and shallow wells and increase the chlorine dose in the water treatment plant and/or introduce or improve the disinfection in the storages along the conventional water distribution system. The last solutions were tested in an experiment that demonstrated that increasing chlorine dosage in drinking water at the water treatment plant reduces the probability of having microbiological contamination at the end user [33].

### 3.2.3. Spreading of Contaminated Water

The high concentration of TC and EC in Group I water was observed in the southern part of the Infulene River Basin. As stated before, the southern part is the area with informal settlements, with limited sanitation, and the water is supplied from a conventional water distribution system with an intermittent supply and poor piping. During rain events, the area is flooded and the contaminants are spread throughout the area and the quality of shallow water might be affected significantly.

## 4. Conclusions and Recommendations

### 4.1. Conclusions

Five points and diffuse sources of pollution were identified in the Infulene River Basin: two wastewater treatment plants, one hospital, one paper manufacturing company, and

one beer factory. Data analysis was performed dividing the set of data in two groups, one related to water in the natural environment and the second dealing with domestic water supplied from different sources in an underserved peri-urban area in a developing world country.

Group I water sources—represented by river water, outlets of wastewater plants, and the drainage system—have been found to present poor results in terms of water quality when comparing most of measured parameters with the standard values. Dissolved oxygen was found to be low in the river water and outlets of wastewater, posing a risk of eutrophication events. Eighty-two (82%) and sixty-one (61%) percent of Group I water samples were positive for TC and EC, respectively. The TC and EC exhibited higher concentrations in the outlets of wastewater treatment plants and drainage waters, with all samples positive, compared to river waters with sixty seven (67%) and twenty seven (27%) percent positive for TC and EC, respectively.

Group II drinking water sources, represented by boreholes, a conventional drinking water distribution system, and shallow water wells, have also been found to have water of poor quality in relation to specific measures parameters threshold values contained in the National Standards for drinking water quality. Observed values of microbial parameters analyzed are TC [541 ($\pm$1523) MPN/100 mL] and EC [43 ($\pm$183) MPN/100 mL].

Sixty three (63%) and twenty-three (23%) percent of Group II drinking water was positive for TC and EC, respectively. Within the group, the borehole and shallow water wells exhibited a considerably higher concentration of TC and EC, compared to the conventional drinking water distribution system water. For samples collected in boreholes, 92% and 38% of samples were positive for TC and EC, respectively; for conventional drinking water distribution systems, 29% and 8% were positive for TC and EC, respectively; and for shallow water wells, 100% of samples were positive for TC and 50% of samples for EC.

*4.2. Recommendations*

Considering the threshold value in the National Standards for water quality parameters for various uses, the river water is good for urban agriculture. However, microbial contamination is an issue that must be considered and impairs its uses for drinking water for example. Further studies require aiming to generate recommendations on how to reduce the risk of use of water for irrigation and on safe handling and consumption of agricultural products from these fields using microbially contaminated water.

Regarding drinking water, studies on the level of residual chlorine at the end users in the conventional drinking water distribution system and how to ensure that there is no microbial contamination before the end user are required. For borehole and shallow well water, a treatment process must be considered to eliminate microbial contamination.

**Author Contributions:** Conceptualization, C.N. and N.C.V.; validation, R.S. and S.M., methods, M.R.; investigation—river water, M.D. and S.R.; investigation—drainage water, H.N.; hydrological modeling, C.M.; investigation—drinking water, D.J. (Dominic Joaquim) and J.L., writing—original draft preparation, C.N. and N.C.V.; writing—review and editing, D.J. (Dinis Juízo), R.S., S.M. and J.S.; supervision, D.J. (Dinis Juízo). All authors have read and agreed to the published version of the manuscript.

**Funding:** This research was funded by the Swedish International Development Cooperation (SIDA) through the Joint Programming Initiative on Antimicrobial Resistance (JPIAMR) program, funding grant number: 274.846C.

**Data Availability Statement:** All data are provided as tables and figures.

**Acknowledgments:** We acknowledge the sharing of experience and support of all colleagues of the consortium that implemented the project on the Surveillance of Emerging Pathogens and Antibiotic Resistances in Aquatic Ecosystems. We also acknowledge the support of the African Population Health Research Center on grant management.

**Conflicts of Interest:** The authors declare no conflict of interest.

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
