# Peer review of "Assessment of Microbial Contamination in the Infulene River Basin, Mozambique"

_water, doi:10.3390/w15020219_

Round 1

Reviewer 1 Report (New Reviewer)

Review of water-1991999: Assessment of microbial contamination in the Infulene River Basin, Mozambique.

The authors conducted a preliminary study on fecal microbial pollution in the drainage system of the Infulene River and nearby Maputo City, as well as an additional investigation on the drinking water used by local community. A comprehensive sampling points 28 sampling points of river water, drainage system and the two outlets of wastewater treatment plants, and 51 drinking water sampling points were selected. A series of physicochemical parameters and microbial contaminants including total coliforms (TC) and escherichia coli (EC) were detected. This study confirmed the high vulnerability to microbial contamination in the system due to poor sanitation and lack of drainage infrastructure. Generally, the manuscript is not well-structured and the presentation is rough. The quality of the manuscript should be further improved. I recommend major revision to this manuscript based on the followed detailed comments.

Introduction:

-       Line 75-84, Authors describe two recent similar studies conducted in Maputo in great detail. And in line 63-65, another similar study was also reported. In this case, what is the novelty and the contribution of this work in the scientific field? Please enhance it in the Introduction section.

-       Authors mainly introduce the recent studies conducted in Mozambique, but an international perspective on the microbial contaminants in rivers is required in this section, not limited to Mozambique or to Maputo.

Methods:

-       Line 102-103, ‘between October to 102 March) and a dry season (April-September)’ is difficult to read and ambiguous. Please make it clear.

-       What is the specific sampling date? Which year? Be specific.

-       Figure 2, there are three maps here but only one scale marked. It is confused. The same with Figure 3.

Results and discussion:

-       In this section, in addition to a description of the research results, a comprehensive discussion of the results should be provided, including a comparison with previous studies, reasons for the results, and potential solutions to microbial contaminants in this area.

-       Line 378-384, Authors give a short description of the spreading of contaminated water, what is the purpose for this section? These sentences are abrupt here. If authors want to explain the causes of microbial contaminants, more specific analysis should be provided.

-       This section is not logic. It is recommended to set up several subtitle to make it clear to read.

Conclusions:

-       This section is too long. Authors needs to summarize all the data of this study and condense into several summative sentences.

Author Response

Reviewer 1

  1. Line 75-84, Authors describe two recent similar studies conducted in Maputo in great detail. And in line 63-65, another similar study was also reported. In this case, what is the novelty and the contribution of this work in the scientific field? Please enhance it in the Introduction section.

Response: Done

  1. Authors mainly introduce the recent studies conducted in Mozambique, but an international perspective on the microbial contaminants in rivers is required in this section, not limited to Mozambique or to Maputo.

Response: Done

Methods:

  1. Line 102-103, ‘between October to 102 March) and a dry season (April-September)’ is difficult to read and ambiguous. Please make it clear.

Response: Improved

  1. What is the specific sampling date? Which year? Be specific.

Response: Done

  1. Figure 2, there are three maps here but only one scale marked. It is confused. The same with Figure 3.

Response: Figure 2 removed; scales added for each map in Figure 3. Numbering of figures updated

Results and discussion:

  1. In this section, in addition to a description of the research results, a comprehensive discussion of the results should be provided, including a comparison with previous studies, reasons for the results, and potential solutions to microbial contaminants in this area.

Response: Improved

  1. Line 378-384, Authors give a short description of the spreading of contaminated water, what is the purpose for this section? These sentences are abrupt here. If authors want to explain the causes of microbial contaminants, more specific analysis should be provided.

Response: Improved

  1. This section is not logic. It is recommended to set up several subtitle to make it clear to read.

Response: Sub-titles included

Conclusions:

  1. This section is too long. Authors needs to summarize all the data of this study and condense into several summative sentences.

Response: This was a bit difficult as the study included drinking water and river water and conclusions on drinking water and surface water are required.

Reviewer 2 Report (Previous Reviewer 2)

I dont think the Fig 2 is meaningful for this paper and propose to delete it.

Author Response

Reviewer 2

  1. I don’t think the Fig 2 is meaningful for this paper and propose to delete it.

Response: Figure 2 removed and figures numbering updated

Round 2

Reviewer 1 Report (New Reviewer)

Accept in present form

This manuscript is a resubmission of an earlier submission. The following is a list of the peer review reports and author responses from that submission.

Round 1

Reviewer 1 Report

The authors tried to highlight the situation of microbial contamination in the river basin water environment. The authors tried to collect and analyzed the physical and chemical parameters of water bodies that are harmful to human health and agricultural production in the various basins of Mozambique. But, there is a serious problem in the methods and result sections. The authors didn't show the relationship between physical and chemical parameters, trends in water quality, and future impact due to changes in water quality changes. This paper's findings showed the regular follow-up data only and did not contain any innovation as well as significant to the future implications.

Introduction

Line no-51: Double commas, please delete one comma.

Materials and Methods

Line no-127: Authors mentioned only sample collection and testing procedure. Please describe the methods of data analysis and interpretation in detail.

Result

I found only descriptive results. Could you please mention the comparison between places, seasonal variation, and its significance to drinking water and agriculture water quality?

Conclusion

Your conclusion is too long. Could you please re-write it in a compact form?

Reviewer 2 Report

General Comments

   To assess the microbial contamination in the Infulene River Basin, Mozambique, this manuscript has tried to apply an integrated study on drainage system, river water, drinking water quality and potentially spreading of microorganisms during the rainy season. However, The World Health Organisation's Guidelines(WHOG) stipulated parameters including physical,chemical, and biological indicators. I would recommend the manuscript include those significant harmful indicators such as persistent organic pollutants (POPs), heave metals etc. In addition, atmospheric input including dry and wet depositions should be considered as a non-point source to bring above mentioned pollutants into the Infulene River Basin.

Special Comments

1. Both Introduction and background were too long and recommended to be combined together each for one or two paragraphs enough.  

2. Sampling and analysis method were very short and suggested to give more detail. 

3. The main parameters or called emphasis such as TC (total coliforms) and EC(escherichia) need to be explained very clearly, especially for EC why which choice escheirichia coliforms?

4. The determination method for TC and EC should be described more detail conformed by the standard method.  

5. line 181 in Figure 3 of page 7, serial number d) should be c).

6. In Figure 8 to 9, all the characters of 1e+09,8e+08, 6e+08, 4e+08,and 2e+08 are present special meaning which should be further explained in more detail.